# Allylic C–H oxygenation of unactivated internal olefins by the Cu/azodiformate catalyst system

Le Wang, Yuan She, Jie Xiao, Zi-Hao Li, Shen-Yuan Zhang, Peng-Fei Lian, Tong-Mei Ding & Shu-Yu Zhang ✉

Allylic ethers and alcohols are essential structural motifs commonly present in natural products and pharmaceuticals. Direct allylic C–H oxygenation of internal alkenes is one of the most direct methods, bypassing the necessity for an allylic leaving group that is needed in the traditional Tsuji–Trost reaction. Herein, we develop an efficient and practical method for synthesizing (E)-allyl ethers from readily available internal alkenes and alcohols or phenols via selective allylic C–H oxidation. Key advances include the use of a Cu/Azodiformate catalyst system to facilitate remote allylic C–H activation and the achievement of excellent chemoselectivity through a dynamic ligand exchange strategy using a bis(sulfonamide) ligand. This method features a broad substrate scope and functional group tolerance, successfully applied to the synthesis of various challenging medium-sized cyclic ethers (7-10 members) and large-ring lactones (14-20 members), with high regioselectivity and stereoselectivity.

Allylic alcohols and ethers are key functional motifs widely found in various bioactive molecules, pharmaceuticals, and natural products, such as aigialomycin, rosuvastatin, and pacritinib (Fig. 1)[1–3]. An attractive alternative involves the direct oxidative coupling of an allylic C–H bond with alcohols or phenols to forge the desired allylic ethers. Compared to the classical Tsuji–Trost substitution reaction, it avoids the prefunctionalization of alkene substrates and exhibits typical features of simplicity in steps, atom economy, and high reaction efficiency[4–6]. However, oxidative allylic etherification is significantly more challenging, as both the substrate alcohols and phenols, as well as the allylic ethers, are prone to competitive oxidation.

In this field, the White group has made significant contributions to both inter- and intramolecular allylic C–H functionalization by developing a Pd(II)/bis-sulfoxide catalytic system[7–14]. This system facilitates the broad cross-coupling of primary, secondary, and tertiary aliphatic alcohols with terminal olefins and can also be employed in the synthesis of various oxygen-containing heterocycles[15,16]. Since then, catalytic systems for oxidative allylic functionalization have developed rapidly.

For instance, Gong and co-workers established a Pd(0)/BQ-P$^{III}$ catalytic system to promote allylic C–H activation, enabling the synthesis of chiral chromans by using a chiral phosphoramidite ligand[17–19]. Recently, the Gevorgyan group reported a photocatalytic platform that operates through a blue light-induced Pd(0/I/II) manifold with a mild aryl bromide oxidant. This method achieves regio- and diastereoselective allylic C–H oxygenation[20–22]. In addition to palladium-based systems, the Blakey group also discovered regioselective intermolecular allylic etherification reactions based on a Rh(III)Cp*/AgOAc-catalyzed platform (Fig. 2a)[23,24]. Despite the significant advances in these methods, they remain limited to simple alkenes with cyclic or terminal double bonds and a narrow range of oxygen nucleophile substrates. Therefore, the development of new catalyst systems capable of promoting regioselective allylic functionalization of internal olefins and compatible with a wide range of nucleophiles is highly desirable.

Recently, we developed an intermolecular allylic C–H amination reaction of internal alkenes, where a π-allyl·copper species (**Int A**) was involved as a key intermediate[25]. Inspired by this study, we turned our

Shanghai Key Laboratory for Molecular Engineering of Chiral Drugs, School of Chemistry and Chemical Engineering, Shanghai Jiao Tong University, Shanghai, PR China. ✉e-mail: zhangsy16@sjtu.edu.cn

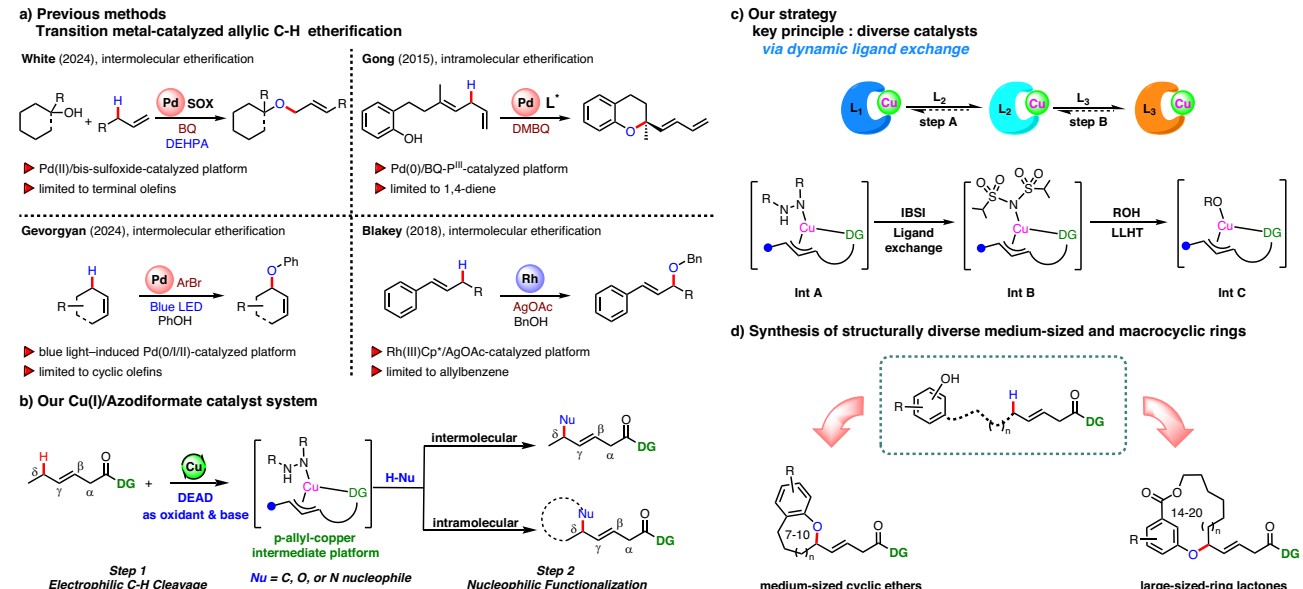

**Fig. 1** | Selected bioactive compounds containing allylic alcohols (Aigialomycin D and Rosuvastatin) and allylic ethers (Herboxidiene, (−)-Zampanolide, (+)-Brasilenyne and Pacritinib).

**Fig. 2** | **Background of current work and reaction design. a** Transition metal-catalyzed allylic C–H etherification. **b** Our Cu(I)/azodiformate catalyst system. **c** Our strategy. **d** Synthesis of structurally diverse medium-sized and macrocyclic rings.

attention to whether this intermediate could be intercepted by a range of stabilized carbon, nitrogen, and oxygen nucleophiles (Fig. 2b). To test the above hypothesis, we unexpectedly found that this catalytic system is suitable for oxygen nucleophiles, with the only drawback being the formation of an allylamine by-product (see the Supplementary Information Table S1 for details). This phenomenon may be attributed to high-basicity nucleophiles, with pKa's ranging from ~10 for phenol to >15 for aliphatic alcohols, which may affect the ligand exchange process and disrupt the catalytic cycle. To overcome this issue, our laboratory has recently disclosed a dynamic ligand exchange strategy to suppress C–N bond reductive elimination (Fig. 2c)[26–31]. Specifically, we introduced the bis(sulfonamide) ligand (**IBSI**) into the reaction system, which rapidly underwent *N*-ligand exchange with **Int A** to form the new π-allyl-Cu complex **Int B**; Subsequently, the hydrogen atom transfers directly from phenol to the *N*-ligand via ligand-to-ligand hydrogen transfer (LLHT)[32–36], forming **Int C**, thereby enabling the formation of mono-allylic etherification product.

Medium-sized cyclic ethers and macrolides are present in numerous biologically important natural products[37,38]. Due to the

unfavorable entropic and enthalpic factors, as well as transannular interactions associated with these ring systems, their efficient synthesis remains a formidable challenge[39–45]. Encouraged by intermolecular etherification reactions, we envisioned that the π-allyl-copper intermediates could be captured by an appropriate intramolecular oxygen nucleophile. As expected, we successfully synthesized a diverse array of medium-sized cyclic ethers (7–10 members) and macrolactones (14–20 members) using our allylic C–H oxidation strategy (Fig. 2d).

## Results

### Reaction condition optimization

After a short survey of potential oxygen nucleophiles, we selected phenol as a suitable starting substrate for our study. We initially tested the standard conditions developed for our recently reported allylic C–H amination reaction but observed an unexpectedly low yield of the desired allylic ether **3a**, along with significant amounts of by-product **3b** (**3a**/**3b** = 23/68). Next, various copper catalysts were examined, including CuCl, CuTc, Cu$_2$O, and CuCl$_2$, and they were less effective than Cu$_2$O (Table 1, entries 1–4). We also investigated polar solvents such as THF, DMF, and the aromatic solvent toluene, which showed no

**Table 1 | Optimization of reaction conditions[a,b]**

| Entry | Cat. (10 mol%) | Ligand (20 mol%) | Pka-(DMSO) | Solvent | 3a-yield (%)[b] | 3b-yield (%)[b] |
|---|---|---|---|---|---|---|
| 1 | CuCl | – | – | DCE | 23 | 68 |
| 2 | CuTc | – | – | DCE | 18 | 71 |
| 3 | Cu₂O | – | – | DCE | 30 | 59 |
| 4 | CuCl₂ | – | – | DCE | 16 | 56 |
| 5 | Cu₂O | – | – | Toluene | 36 | 45 |
| 6 | Cu₂O | – | – | DMF | 15 | 67 |
| 7 | Cu₂O | – | – | THF | 23 | 51 |
| 8 | Cu₂O | L1 | 5.11 | Toluene | nr | nr |
| 9 | Cu₂O | L2 | 8.08 | Toluene | 34 | trace |
| 10 | Cu₂O | L3 | 8.60 | Toluene | 46 | trace |
| 11 | Cu₂O | L4 | 8.82 | Toluene | 68 | 12 |
| 12 | Cu₂O | L5 | 9.82 | Toluene | 61 | 20 |
| 13 | Cu₂O | L6 | 11.38 | Toluene | 41 | 42 |
| 14 | Cu₂O | L7 | 12.60 | Toluene | 36 | 43 |
| 15 | Cu₂O | L8 | 9.24 | Toluene | 67 | 12 |
| 16 | Cu₂O | L9 | 9.97 | Toluene | 88 | trace |
| 17 | Cu₂O | L10 | 9.26 | Toluene | 42 | trace |
| 18 | Cu₂O | L11 | 9.05 | Toluene | nr | trace |

[a]Reaction conditions: **1a** (0.2 mmol, 1.0 equiv), **2a** (0.6 mmol, 3.0 equiv), DEAD (0.4 mmol, 2.0 equiv), catalyst (10 mol%) and ligand (20 mol%) in solvent (2.0 mL) at the 80 °C for 10 h.
[b]Isolated yield. See Supplementary Information for more conditions.

significant impact on both yield and chemoselectivity (Table 1, entries 5–7). Inspired by ligand exchange strategies, we hypothesized that introducing a protonic acid into the reaction system might undergo proton transfer with azodicarboxylate, thereby facilitating its dissociation from the copper catalyst as hydrazinedicarboxylate. Subsequently, various protonic acids were screened as ligands (see the Supplementary Information Table S4 for details). The results indicate that as the acidity of the ligand decreased, only the desired compound **3a** was produced, with the yield gradually increasing from 0% to 68% (Table 1, entries 8–11). However, further reduction in ligand acidity resulted in a decrease in the chemoselectivity of the product, approaching a ratio of 3:4 (Table 1, entries 12–14). To our delight, the choice of ligand **BBSI** had a remarkable impact on the chemoselectivity and exhibited much better performance, giving **3a** in 68% yield, while the side product **3b** was observed in 12% yield (**3a/3b** = 68/12) (Table 1, entry 11). It can be inferred that the acidity of the ligand significantly influences the chemoselectivity of the product. Weaker acidity affects the *N*-ligand exchange process, while stronger acidity impacts the LLHT process. Encouraged by these results, we evaluated **BBSI** with different substituents (Table 1, entries 15–18) and found that

iPr-**L9** (**IBSI**) exhibited the best performance, furnishing **3a** in 88% yield with complete inhibition of the side product **3b**. Notably, **L11**, which has methyl groups at positions 1, 3, and 5 of the benzene ring, exhibited no reactivity, potentially attributed to steric hindrance preventing the LLHT process (more details, see Supplementary Information Table S5).

## Substrate scope

Under optimized conditions, the substrate scope of the δ-etherification reaction was investigated (Fig. 3). In all cases, the reaction proceeded with high regioselectivity and excellent *E/Z* selectivity (>20:1). Firstly, we explored the scope of alkene substrates using phenol and found that the reaction was compatible with a variety of substituted internal alkenes. δ-Etherification with linear, branched, and cyclic substituents resulted in moderate to excellent yields (**3a**–**9**). Alkene substrates bearing a di-α-substituent were suitable for this reaction, while mono-α-substituted substrate underwent this transformation with good yield and a 1:1 dr (**4**, **5**). Interestingly, we further evaluated the impact of long alkyl chain substituents with various functional groups. Substituents including chlorine, iodine, ether, ester,

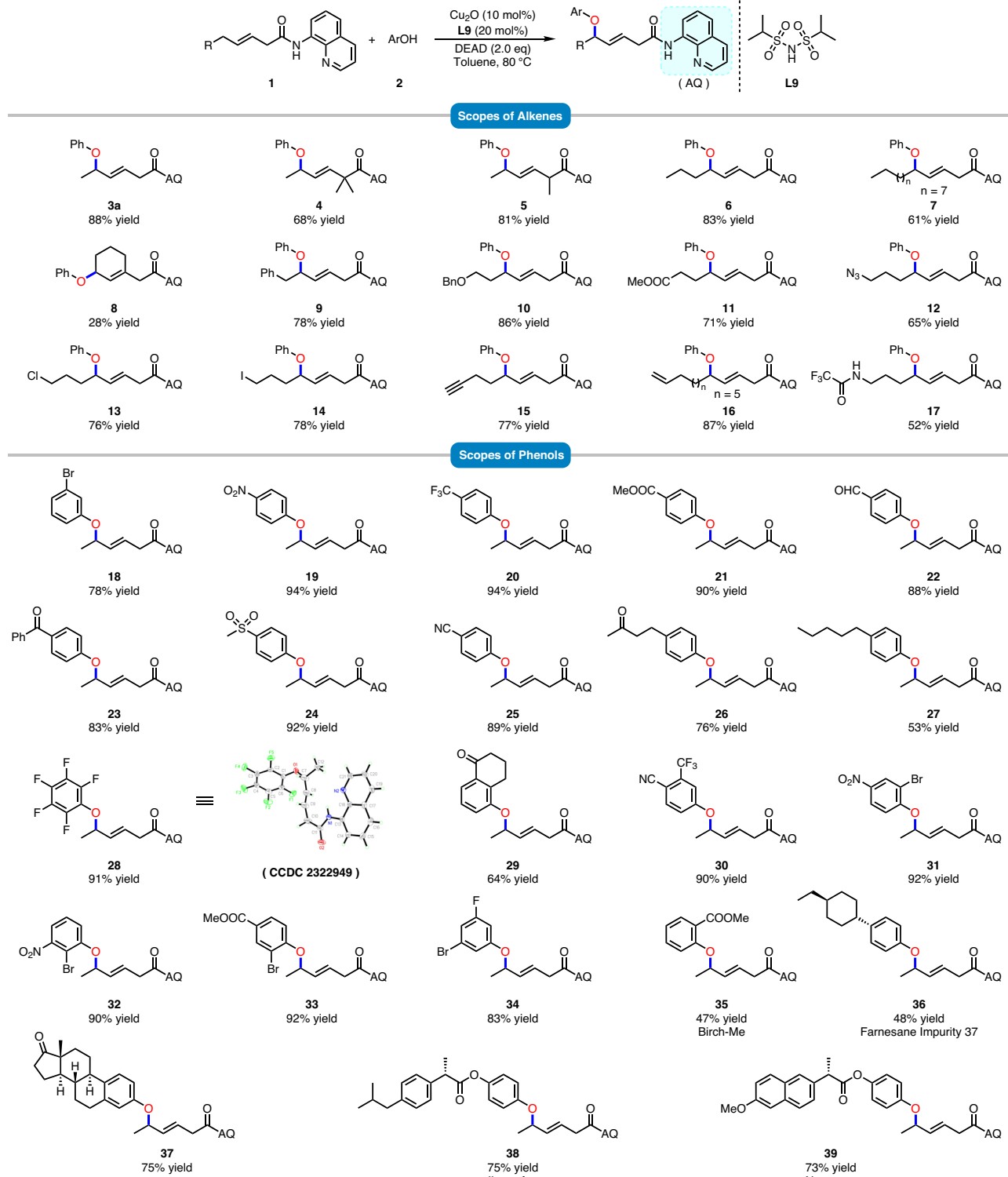

**Fig. 3 | Substrate scope of various olefins and phenols[a,b]. a** Reaction conditions: **1a** (0.2 mmol, 1.0 equiv), **2** (0.6 mmol, 3.0 equiv), DEAD (0.4 mmol, 2.0 equiv), $Cu_2O$ (10 mol%) and **L9** (20 mol%) in toluene (2.0 mL) at the 80 °C for 10 h. **b** Isolated yield.

azide, and amide were all well tolerated (**10–17**). In addition, substrates containing terminal alkyne or terminal alkene moieties were also applicable to this reaction (**15, 16**). This insight was then implemented in reactions with numerous phenol coupling partners. A diverse range of electron-withdrawing and electron-donating groups at various positions on the aryl units were found to be well tolerated, resulting in the corresponding products with yields ranging from 53% to 94%

(**18–28**). The molecular structure of compound **28** has been confirmed by X-ray crystallography. For multisubstituted phenols (**29–34**), the reactions also proceeded smoothly to afford the corresponding products in satisfactory yields. Additionally, a variety of functional groups, including halogen (**18, 28, 34**), nitro (**19, 31, 32**), aldehyde (**22**), trifluoromethyl (**20, 30**), sulfonyl (**24**), carboxylic ester (**21, 33**), ketone (**23, 26, 29**), and nitrile (**25, 30**), were well tolerated on the aryl ring

**Fig. 4 | Substrate scopes of various alcohols[a,b]. a** Reaction conditions: **1a** (0.2 mmol, 1.0 equiv), **2** (0.6 mmol, 3.0 equiv), DEAD (0.4 mmol, 2.0 equiv), CuCl (10 mol%) and **L12** (20 mol%) in toluene (2.0 mL) at the 80 °C for 10 h. **b** Isolated yield.

under standard conditions. The etherification reaction also proceeded effectively in the late-stage functionalization of several pharmaceuticals and related bioactive molecules (**35–39**), including the cosmetic ingredient Birch-Me, the pharmaceutical molecule Farnesane Impurity 37, the natural product estrone, as well as analogs of ibuprofen and naproxen.

To further demonstrate the synthetic potential of this transformation, various alcohols were subsequently introduced into the etherification reaction (Fig. 4). Under the optimal conditions for phenol, a small amount of compound **41** was detected. By referencing Stahl's work, modifying the reaction conditions to the CuCl/(EtO)$_2$P(O)H catalytic system significantly improved the yield of the etherification product[46]. The substrates of methanol (**40**), ethanol (**41**), cyclopropylmethanol (**42**), benzylalcohol (**43, 44**), and 3-phenylpropanol (**45**) were all allowed to react with **1a**, furnishing the corresponding products in moderate yields. Several ethanol derivatives with various substituents, including bromine, ester, thiophene, alkynyl, vinyl, and naphthyl groups (**46–53**), exhibit favorable performance in this reaction, affording satisfactory yields in most cases. From products **54–57**, it can be observed that alkyl alcohols with trifluoromethyl substitution, as the chain length increases, exhibit a corresponding decrease in product yields. This suggests that the reaction rate may be related to the ease of deprotonation of the O-nucleophilic reagent.

Having successfully achieved intermolecular etherification, we hypothesized whether this catalytic mode could be extended to intramolecular C–H oxidation reactions for the construction of cyclic compounds (Fig. 5). Initially, we attempted the synthesis of more synthetically favorable 5-membered and 6-membered ring compounds. The results indicated that we could obtain tetrahydrofuran (**58**), tetrahydropyran (**59, 60**), benzofuran (**61**), and benzopyran (**62, 63**) derivatives in moderate yields. It is gratifying to note that the current strategy enables the successful synthesis of medium-sized rings, which are typically considered difficult to prepare. Substrates bearing trifluoromethyl electron-withdrawing groups smoothly obtained the corresponding 7- to 10-membered cyclic ether products (**64–67**). Subsequently, by employing the different chain lengths of alkyl groups, all 14- to 20-membered macrolides (**68–74**) can be smoothly obtained in 53–78% yields.

## Synthetic applicability

To demonstrate the applicability and effectiveness of our strategy, both intermolecular and intramolecular gram-scale reactions were conducted under standard conditions. The reactions afforded product **3a** in 82% yield (2.72 g) and product **70** in 62% yield (1.51 g) (Fig. 6a). The internal alkenes in the products can be reduced to form **75** or oxidized to generate **76** using K$_2$O$_s$O$_4$. Reduction with LiAlH$_4$ transformed the amide moiety into the corresponding secondary amine to furnish **77**, while the 8-aminoquinoline moiety could be conveniently removed by treatment with IBX, affording the primary amine **78** (Fig. 6b). Subsequently, compound **19** was subjected to indium-mediated reduction of the nitro group, followed by CAN oxidative cleavage to afford the linear allylic alcohol **79** (Fig. 6c), which can serve as an important synthetic intermediate.

## Mechanistic study

To investigate the underlying mechanism of the allylic C–H etherification process, a series of control and isotope labeling experiments were conducted. Initially, it was observed that both trans- and cis-hexenamides (**1a** and **80**) produce the same product **3a**, indicating that the reaction proceeds via a π-allyl-Cu intermediate (Fig. 7a). No reaction was observed with the N-methyl protected enamide substrate **81**, highlighting the critical chelating role of the directing group in the reaction (Fig. 7b). In the presence of two equivalents of TEMPO or BHT as radical scavengers, **3a** could still be isolated in 82% and 78% yields, respectively. These results suggest that the reaction may not proceed via a radical pathway (Fig. 7c). Next, the by-product **3b** was selected as the substrate and reacted under standard conditions, but no etherification product was obtained. This suggests that the reaction is unlikely to proceed via a nucleophilic substitution mechanism (Fig. 7d). In addition, a deuterium labeling experiment with **83-d$_2$** (59% D) produced **86-d$_2$** (59% D) in an 83% yield without deuterium scrambling and the deuteration at the internal alkene of **84-d$_2$** (97% D) turned out to be identical to that of the corresponding product **87-d$_2$** (97% D), which strongly supported that the alkene isomerization of **1a** would not occur during the reaction process. Moreover, the reaction of **85-d$_3$** (>99% D) and 4-nitrophenol smoothly afforded the product **88-d$_2$** with 98% deuteration at the allylic position, suggesting that the allylic C–H cleavage process was most likely to be irreversible

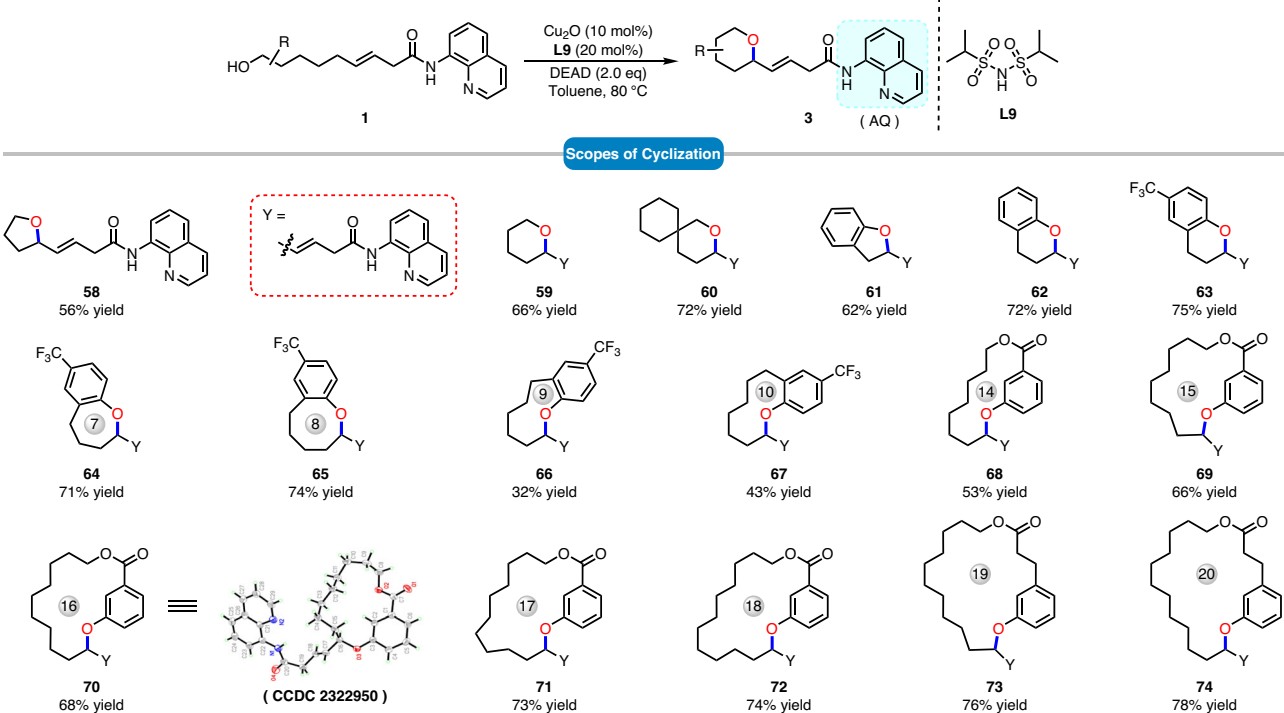

**Fig. 5 | Substrate scopes of intramolecular reactions$^{a,b}$. a** Reaction conditions: **1** (0.2 mmol, 1.0 equiv), DEAD (0.4 mmol, 2.0 equiv), Cu$_2$O (10 mol%) and **L9** (20 mol%) in toluene (2.0 mL) at the 80 °C for 10 h. **b** Isolated yield.

(Fig. 7e). Furthermore, the parallel and intermolecular kinetic isotope effects were determined to be 2.5 and 2.3, respectively (Fig. 7f, g). All these results are consistent with the allylic C–H activation pathway, indicating that the allylic C–H cleavage likely involves the rate-limiting step.

### DFT calculations
To gain a better understanding of the reaction mechanism, particularly the crucial role of ligand exchange strategies in determining the reaction's chemoselectivity, density functional theory (DFT) calculations were conducted (Fig. 8). The computational study on the Cu/azodiformate system-catalyzed allylic C–H activation, leading to the formation of the π-allyl·Cu complex **A**, can be referenced from our previous work[25]. From the complex **A**, in the absence of a protic acid ligand, the reaction more readily proceeds through **TS1-1** to undergo C–N bond reductive elimination, forming intermediate **B-1** with a Gibbs free-energy barrier of 7.6 kcal/mol. However, upon the introduction of the **IBSI**, intermediate **A** rapidly undergoes ligand exchange with **IBSI** via **TS1**, forming intermediate **C** with a Gibbs free-energy barrier of 5.9 kcal/mol. For acetic acid and phenol, the protonation transition states **TS-1'** and **TS-1''** exhibit relatively high energies, indicating that these ligands are less prone to undergo proton transfer. This suggests that the driving force for ligand exchange is related to the acidity of the ligands. Moreover, intermediate **C** faces challenges with C–N bond reductive elimination due to the high energy barrier of **TS-2'**. Instead, it undergoes LLHT with phenol, transitioning via **TS2** and **TS3** successively, forming intermediate **F**. This intermediate then dissociates the **IBSI** to produce the new π-allyl-copper species **G**, which is capable of undergoing C–O bond reductive elimination (**TS4**) to generate the desired product. Regarding regioselectivity, DFT calculations were employed to investigate two competing transition states in the inner-sphere pathway. The δ-amination pathway through **TS4** was found to be at least 8.8 kcal/mol more favorable than the β-amination pathway.

### Discussion
In summary, we have developed a general Cu/azodiformate system-catalyzed allylic oxidation method that efficiently affords (E)-allyl-lethers from readily available internal olefins and alcohols or phenols with various functional groups, exhibiting high regioselectivity and stereoselectivity. This approach allows for the efficient synthesis of a variety of medium-sized cyclic ethers (7–10 members) and large-ring lactones (14–20 members). Experimental and computational studies indicated how an unusual mechanism involving dynamic ligand exchange can be used to control the chemoselectivity and ultimately expand C–O bond-forming methodologies. Insights from this study will help broaden the scope of pro-nucleophiles used in allylic C–H functionalization reactions.

### Methods
#### General procedure
A mixture of amide (0.20 mmol, 1.0 equiv), Cu$_2$O (0.02 mmol, 0.1 equiv), L (0.04 mmol, 0.2 equiv), DEAD (0.40 mmol, 2.0 equiv) and alcohol/phenol (0.60 mmol, 3.0 equiv) in Toluene (2 mL) in a 10 mL glass vial (sealed with PTFE cap) was heated at 80 °C for indicated time. The reaction progress was monitored by thin-layer chromatography. Upon completion, the reaction mixture was concentrated in vacuo and purified by silica gel column chromatography to afford the desired products.

### Data availability
The data generated in this study are provided in the Supplementary Information file. The experimental procedures, data of NMR, and HRMS have been deposited in the Supplementary Information file. The X-ray crystallographic coordinates for structures reported in this study have been deposited at the Cambridge Crystallographic Data Centre (CCDC: 2322949) and (CCDC: 2322950). These data could be obtained free of charge from The Cambridge Crystallographic Data Centre (https://www.ccdc.cam.ac.uk/data_request/cif). All data are available

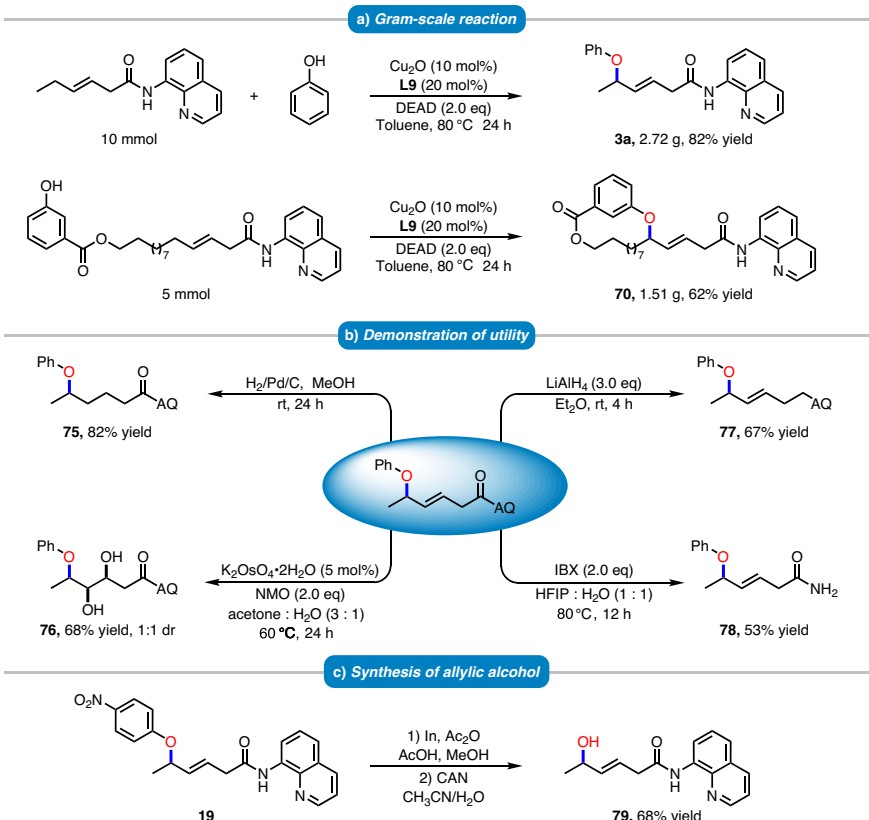

**Fig. 6 | Synthetic applicability. a** Gram-scale reaction. **b** Demonstration of utility. **c** Synthesis of allylic alcohol.

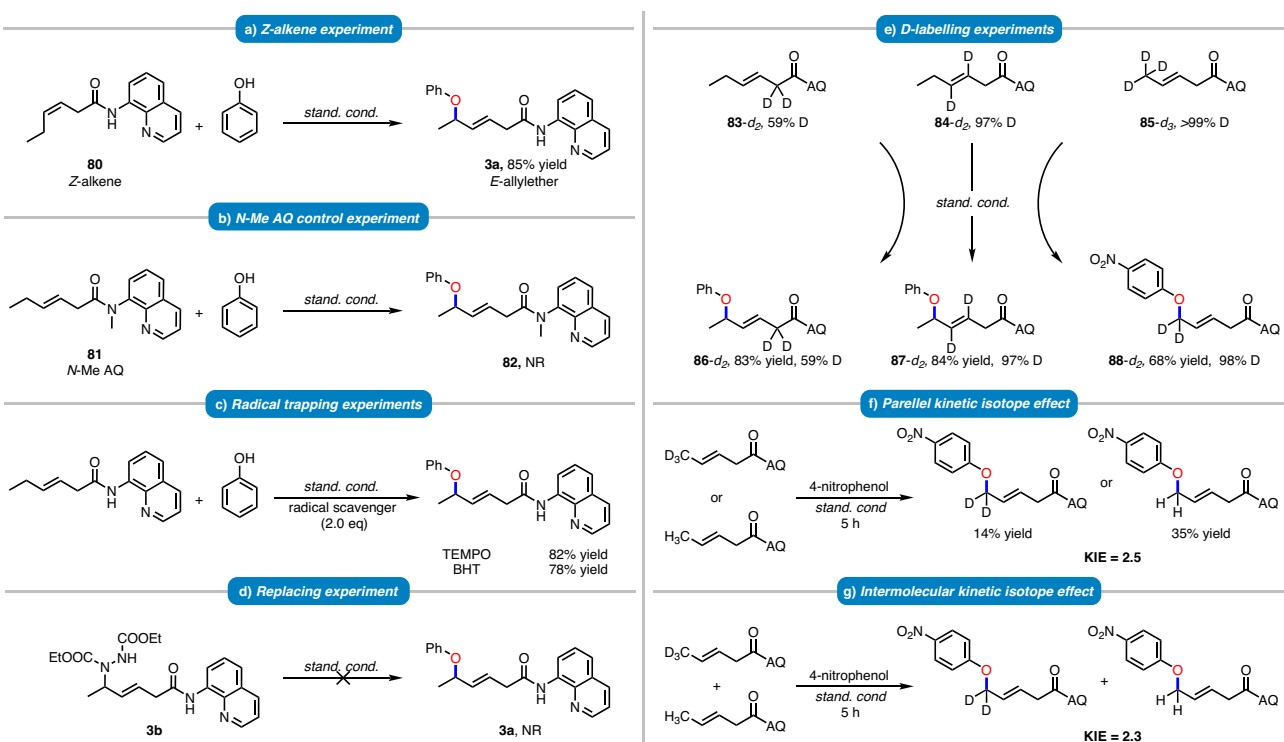

**Fig. 7 | Mechanistic experiment. a** *Z*-alkene experiment. **b** N-Me AQ control experiment. **c** Radical trapping experiments. **d** Replacing experiment. **e** D-labeling experiment. **f** Parellel kinetic isotope effect. **g** Intermolecular kinetic isotope effect.

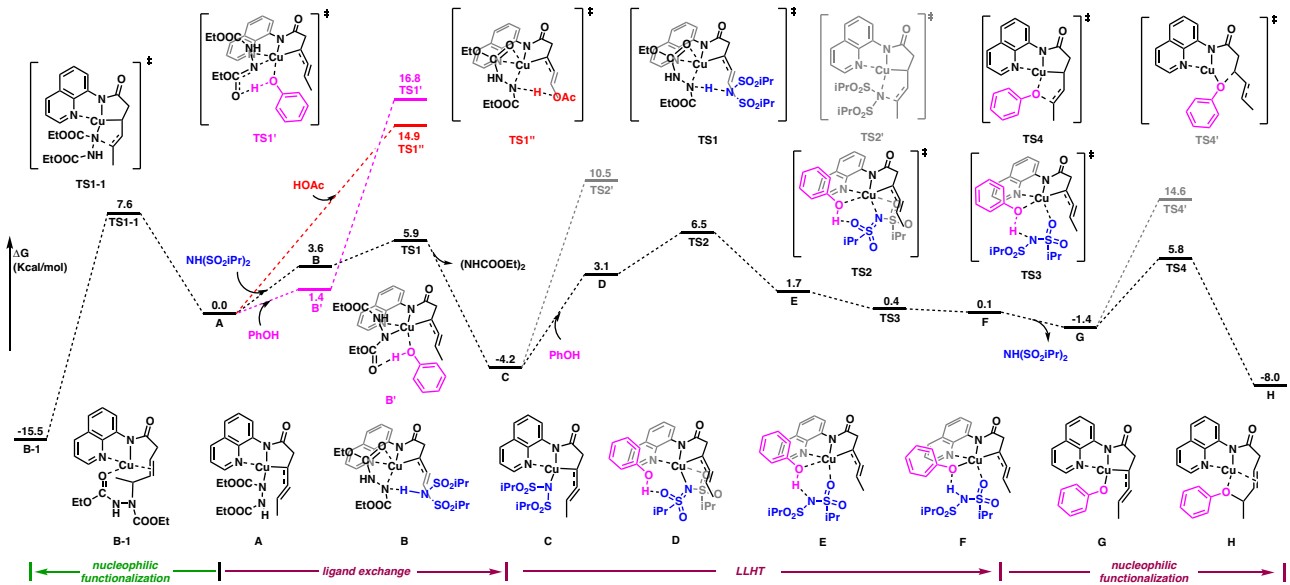

**Fig. 8 |** The DFT-computed free-energy changes of the favorable reaction pathway and the Gibbs free energies of the key transition states.

from the corresponding author upon request. Source data are provided with this paper.

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

## Acknowledgements

This work was supported by the National Key Research & Development Program of China (2021YFF0701800, S.-Y.Z. and 2022YFC2703400, S.-Y.Z.), the Fundamental Research Funds for the Central Universities (YG2022ZD021, S.-Y.Z.), and STCSM (22ZR1435100, S.-Y.Z.).

## Author contributions

The project was conceived and directed by S.-Y.Z. (Shu-Yu Zhang), L.W. designed the experiments and analyzed the data. S.-Y.Z. (Shen-Yuan Zhang) and Z.-H.L. performed the DFT calculations. L.W., Y.S., and J.X. performed the experiments. L.W., P.-F. L., and T.-M. D. prepared the manuscript. All authors discussed the results and commented on the manuscript.

## Competing interests

The authors declare no competing interests.
