## [Transparent Peer Review file · Nature Communications]

Allylic C–H Oxygenation of Unactivated Internal Olefins by the Cu/Azodiformate Catalyst System

Corresponding Author: Professor Shuyu Zhang

Version 0:

Reviewer comments:

Reviewer #1

(Remarks to the Author)

This manuscript presents a general method for Cu/azodiformate-catalyzed allylic oxidation, efficiently converting readily available internal olefins and alcohols or phenols with various functional groups into (E)-allyl ethers. While the applicability of this method for synthesizing macrocyclic lactones is appealing, the novelty is somewhat diminished by the authors' previous related work (ref 25). The current work builds upon their previous allylic amination study (Nat. Commun. 2024, 15, 1483) where DEAD was used as an amine source. In this approach, they suppress DEAD's direct reductive elimination through ligand exchange and introduce additional nucleophiles to achieve allylic oxygenation. Although this reaction still requires a bidentate directing group, it shows utility in several aspects: Compatibility with various alcohols, applicability to both internal and terminal alkenes, and a new approach to suppressing existing reactivity through rapid ligand exchange.

Specific Comments

-Oxidant Screening: Have you conducted a comprehensive screening of oxidants? Using alternative oxidants might eliminate concerns about side products.

-Directing Group Specificity: Is AQ the only effective directing group? Please clarify if other directing groups, including monodentate ones, were tested.

-Ligand Selection Rationale: Please explain why L12 is necessary for intermolecular etherification (Table 3) while L9 suffices for intramolecular etherification (Table 4).

-Correction: Line 106 should reference entries 12-14.

-Scope Demonstration: Figure 1 should include at least one example containing both an allylic carbonyl and an allyl ether group to adequately demonstrate the method's applicability, given the reactive ion considerations.

-Including analysis of the deuterated hydrazine by-product would strengthen the mechanistic arguments (as compared to ref. 25).

-Reference 26 does not properly support the discussion of C–N bond reductive elimination suppression through ligand exchange.

Reviewer #2

(Remarks to the Author)

The manuscript authored by Zhang et al. introduces an innovative strategy for the oxidative intermolecular amination of allylic C–H bonds, marking a significant leap forward in organic synthesis, especially concerning the synthesis of aliphatic allylic amines, which are ubiquitous in natural products and pharmaceuticals. This research timely tackles a formidable

challenge in synthetic chemistry: the selective amination of unactivated internal alkenes. The authors have masterfully crafted a Cu-catalyzed approach that employs azodiformates in a dual capacity as a nitrogen source and an electrophilic oxidant, a resourceful strategy that surmounts the conventional hurdles encountered in such transformations. The manuscript is commendably well-structured, with results articulated in a clear and methodical manner, accompanied by an exhaustive series of experiments that delineate the method's scope and constraints. Consequently, I strongly recommend the publication of this manuscript, albeit with the following two minor recommendations for further enhancement:

1. The manuscript would benefit from a careful proofreading to correct minor formatting and typographical errors. For instance, in Table 1, there should be a space between numerical values and the unit "mol%" (e.g., "10 mol%" instead of "10mol%"). Additionally, the abbreviation for isopropyl in the Table of Contents (TOC) and Figure 5 is inconsistent and should be standardized to "-Pr" throughout the document. These corrections are essential to maintain the professional appearance of the manuscript and to avoid any confusion for the readers.

2. One of the aspects that would greatly enhance the innovation of this work is the demonstration of stereoselective control in the reaction, particularly for alkyl alcohol substrates. The use of chiral phosphite ligands could potentially impart enantioselectivity to the reaction, which would be a significant advancement. I recommend that the authors explore this possibility in a follow-up study or, if feasible, include some preliminary data in the current manuscript. The successful implementation of such a strategy would not only broaden the applicability of the developed method but also provide a deeper understanding of the reaction mechanism.

Reviewer #3

(Remarks to the Author)

Zhang and co-workers have developed a Cu/Azodiformate system-catalyzed allylic oxidation method that efficiently affords (E)-allylethers from internal olefins with the direct group and alcohols or phenols, exhibiting high regioselectivity and stereoselectivity. This approach enables the synthesis of a variety of medium-sized cyclic ethers and large-ring lactones. Experimental and computational studies indicated that how a distinguished mechanism involving dynamic ligand exchange can be used to control the chemoselectivity and ultimately expand C–O bond-forming methodologies. The manuscript is nearly ready for publication but requires final minor revisions before it can be accepted.

1. In the TOC and Figure 2b-c, the structures with square brackets and double dagger are corresponding to the transition state, not the intermediates. Please correct it.
2. As the allyl alcohol moiety has been mentioned in Aigialomycin D and Rosuvastatin in Figure 1, and the multistep reaction for the synthesis of allyl alcohol has been conducted and shown in Figure 3c, is there any effort to directly synthesize allyl alcohol in one step?
3. The Pka of ligands in DMSO has been displayed in Table 1, it seems that the relevant discussion is missing in the text.
4. For substrate scopes of alcohols, it is better to provide the reactivity of secondary alcohols and tertiary alcohols.
5. For the DFT calculations, has any conformation search been performed for the intermediates and transition states to provide the most stable points? For instance, have the more stable conformation of A and the other conformation of TS1' including proton-transfer from O-H-N analogous to TS1 with proton-transfer from N-H-N been located? From C to D in LLHT, how does the Cu-N bond break before proton-transfer? Finally, G is endoergic from C and the free energy barrier is not so high, indicating that the process seems to be a reversible reaction. Is there any good discussion? By the way, the unit of G in Figure 5 should be kcal/mol.
6. In Ref. 47, the abbreviation of Nature Catalysis should be given.
7. In P91 in SI, the integration was in casual version in the ¹H NMR spectra, and the chemical shift was incorrect in the ¹³C NMR spectra, please standardize the handling of NMR spectra. In P93, 94, 97, 101, 114, 137, 167, 168 in SI, the integration in the ¹H NMR spectra. In P108, 127 in SI, the compounds should be further purified. Please add the compound number in the SI for better reading.

Reviewer #4

(Remarks to the Author)

Zhang group develops an allylic C-O coupling between internal alkenes and alcohols through remote directing strategy involving a crucial dynamic ligand exchange, based on their recent work on a similar allylic C-N coupling (Ref. 25). The reaction conditions have been systematically optimized, the scope of alkenes and alcohols have been thoroughly explored, and the mechanism has also been comprehensively studied, both experimentally and theoretically. This is a solid study with good novelty and remarkable value. I would hereby strongly recommend accepting this work for publication on Nat. Commun. after minor revisions.

(1) The abstract could be condensed. My inclination is that the abstract should focus on description of the content of this study (what has been done and what can they demonstrate), and the significance of this study is not necessarily involved in the abstract. Thus the first two sentence "Allylic ethers and alcohols..... Tsuji-Trost reaction" should better be deleted, and detailed discussion on the significance of this work should better be included in the Introduction section.

(2) According to the results shown in Table 2, electron-withdrawing substitutions are well tolerated (even preferable) but weak electron-donating alkyl or acyloxy substitutions seem less tolerable (see product 26, 27, 38 and 39). How about strong electron-donating alkoxy (such as p-methoxyphenol), hydroxy (such as quinol) or amino/amido (such as 4-acetamidophenol, a.k.a. Paracetamol, one of the most representative NSAIDs beside Ibuprofen) substitutions? I also wonder why the authors select a model phenol with a long alkyl chain (see product 27). Can this significant C-O coupling takes

place with the most typical alkyl phenol, p-cresol?

(3) According to the results shown in Table 3, a series of primary alcohol could smoothly react albeit in declined yields. How about secondary alcohols (such as isopropanol) beside HFIP that is too specific?

(4) This remote directed ligand exchange strategy to facilitate allylic C-O coupling of unactivated alkenes are very significant and promising. However, with regards to application and utility, the amide linker between olefin and the quinoline directing group might not be the most satisfactory solution. According to the results shown in Figure 3b, the effectiveness for removal of the DG is not very well, since an excessive amount of a relatively expensive oxidant IBX is employed and the 53% yield is too low for utility consideration. It is strongly proposed that, in future studies, the authors develop more applicable linkers other than amide bond, as well as more atom-economical DGs other than quinoline.

Version 1:

Reviewer comments:

Reviewer #1

(Remarks to the Author)

The authors have addressed the concerns and questions and the revised version can now be acceptable.

Reviewer #3

(Remarks to the Author)

I recommend the publication of this manuscript now.

Reviewer #4

(Remarks to the Author)

All issues indicated by this and other referees have been well-addressed. I therefore recommend accepting this manuscript for publication on Nat. Commun. in its current form.

List of responses

The main corrections in the paper and the responds to the reviewers' comments are as follows:

1. Responses to reviewer 1:

1) Comment #1: "Oxidant Screening: Have you conducted a comprehensive screening of oxidants? Using alternative oxidants might eliminate concerns about side products."

Our Response: Thanks for your suggestions. We have conducted extensive screening of oxidants, and the results show that these oxidants not only failed to eliminate byproducts but also caused the reaction to cease, further demonstrating the unique properties of azodiformate as an oxidant.

2) Comment #2: "Directing Group Specificity: Is AQ the only effective directing group? Please clarify if other directing groups, including monodentate ones, were tested."

Our Response: Thanks for your suggestions. We have screened a variety of directing groups, as shown in the figure below. The results indicate that AQ and its substituted derivatives efficiently lead to the desired product, while other directing groups, including those with bidentate auxiliary groups (**3e**) and monodentate auxiliary groups (**3f**, **3g**), failed to yield the target product. Only a small amount of product was detected when using the 2-(pyridin-2-yl)isopropyl (PIP) directing group (**3d**).

3) Comment #3: "Ligand Selection Rationale: Please explain why **L12** is necessary for intermolecular etherification (Table 3) while **L9** suffices for intramolecular etherification (Table 4)."

Our Response: Thanks for your suggestions. The ligand screening results indicate that ligand **L12** is favorable for the coupling of alkyl alcohol substrates, while ligand **L9** is advantageous for the coupling of phenol substrates. Based on these results, we hypothesize that since alkyl alcohols are less acidic than phenols, the ligands involved in the **LLHT** process must match the acidity of the substrates for effective ligand exchange. This may require different ligands to undergo exchange with nucleophilic oxygen reagents of varying acidities.

4) Comment #4: “Line 106 should reference entries 12-14.”

Our Response: Thanks for your suggestions. We have made the correction in the manuscript.

5) Comment #5: “Scope Demonstration: Figure 1 should include at least one example containing both an allylic carbonyl and an allyl ether group to adequately demonstrate the method’s applicability, given the reactive ion considerations.”

Our Response: Thanks for your suggestions. We conducted extensive searches across multiple databases and found it challenging to identify bioactive molecules or natural products that simultaneously contain both allylic carbonyl and allyl ether groups. Most of the bioactive molecules we found were α,β -unsaturated compounds with ether groups at the allylic position.

6) Comment #6: “Including analysis of the deuterated hydrazine by-product would strengthen the mechanistic arguments (as compared to ref. 25).”

Our Response: Thanks for your suggestions. In reference 25, we have thoroughly investigated the mechanism of allylic C-H bond cleavage mediated by a Cu/azodiformate catalytic system, which ultimately leads to the formation of deuterated allyl hydrazine as a product. For the allylic etherification reaction, the mechanistic arguments for the cleavage of the allylic C-H bond can be referenced from ref. 25. However, when the **IBSI** ligand is introduced into the system, it becomes difficult to obtain the allyl hydrazine by-product in this catalytic system due to the occurrence of *N*-ligand exchange.

7) Comment #7: “Reference 26 does not properly support the discussion of C–N bond reductive elimination suppression through ligand exchange.”

Our Response: Thanks for your suggestions. We have removed reference 26 from the manuscript.

2. Responses to reviewer 2:

1) Comment #1: “The manuscript would benefit from a careful proofreading to correct minor formatting and typographical errors. For instance, in Table 1, there should be a space between numerical values and the unit “mol%” (e.g., “10 mol%” instead of “10mol%”). Additionally, the abbreviation for isopropyl in the Table of Contents (TOC) and Figure 5 is inconsistent and should be standardized to “*i*-Pr” throughout the document. These corrections are essential to maintain the professional appearance of the manuscript and to avoid any confusion for the readers.”

Our Response: Thanks for your suggestions. We have carefully proofread the manuscript, corrected formatting and typographical errors, and standardized the abbreviation for isopropyl in the Table of Contents (TOC) and Figure 5.

2) Comment #2: “One of the aspects that would greatly enhance the innovation of this work is the demonstration of stereoselective control in the reaction, particularly for alkyl alcohol substrates. The use of chiral phosphite ligands could potentially impart enantioselectivity to the reaction, which would be a significant advancement. I recommend that the authors explore this possibility in a follow-up study or, if feasible, include some preliminary data in the current

manuscript. The successful implementation of such a strategy would not only broaden the applicability of the developed method but also provide a deeper understanding of the reaction mechanism.”

Our Response: Thanks for your suggestions. We have previously used chiral phosphite ligands to construct chiral C-O bonds, but they did not impart enantioselectivity to the reaction. Currently, research on chiral catalytic versions of this reaction is ongoing.

3. Responses to reviewer 3:

1) Comment #1: “In the TOC and Figure 2b-c, the structures with square brackets and double dagger are corresponding to the transition state, not the intermediates. Please correct it.”

Our Response: Thanks for your suggestions. We have made the correction by removing the double dagger and retaining only the square brackets.

2) Comment #2: “As the allyl alcohol moiety has been mentioned in Aigialomycin D and Rosuvastatin in Figure 1, and the multistep reaction for the synthesis of allyl alcohol has been conducted and shown in Figure 3c, is there any effort to directly synthesize allyl alcohol in one step?”

Our Response: Thanks for your suggestions. We previously attempted to use water as a nucleophile for the one-step synthesis of allyl alcohol but did not obtain the desired product. Ongoing research is being conducted to address this issue.

3) Comment #3: “The Pka of ligands in DMSO has been displayed in Table 1, it seems that the relevant discussion is missing in the text.”

Our Response: Thanks for your suggestions. We have added a discussion on the impact of the ligand's pKa on the chemoselectivity of the reaction in the "Reaction Condition Optimization" section of the manuscript.

4) Comment #4: “For substrate scopes of alcohols, it is better to provide the reactivity of secondary alcohols and tertiary alcohols.”

Our Response: Thanks for your suggestions. We have previously investigated the reactivity of secondary and tertiary alcohols, but the results were not optimal. Due to constraints on the presentation of substrates in the manuscript, we chose not to include these findings in the main text and instead placed them in the Supplementary Information.

5) Comment #5: “For the DFT calculations, has any conformation search been performed for the intermediates and transition states to provide the most stable points? For instance, have the more stable conformation of A and the other conformation of TS1’ including proton-transfer from O-H-N analogous to TS1 with proton-transfer from N-H-N been located? From C to D in LLHT, how does the Cu-N bond break before proton-transfer? Finally, G is endoergic from C and the free energy barrier is not so high, indicating that the process seems to be a reversible reaction. Is there any good discussion? By the way, the unit of ΔG in Figure 5 should be kcal/mol.”

Our Response: Thanks for your suggestions. First, we have provided a new calculation

pathway and discussion structures with conformational search using CREST with XTB and manual attempts. Additionally, we have added TS2 in the process from C to D to describe the breakage of the Cu-N bond.

Second, for A, after conformational search, we did find a lower-energy structure, with the main difference being the orientation of the protonated group in DEAD. However, the corresponding transition state (TS1-new) has a higher energy. Therefore, we have still adopted the previous A structure, as shown in Figure 1. Additionally, we have also explored a new TS1' with an O-H-N proton transfer structure, as shown in Figure 2.

Finally, for the process from C to G (new H), new energy values were obtained after the conformational search. Additionally, due to the length limitation of the figure, we did not include the subsequent transformation process in the figure. As shown in Figure 3, the further transformation will lead to an additional decrease in the system's energy. We will include this part in the Supplementary Information.

Figure 1

Figure 2

Figure 3

6) Comment #6: “In Ref. 47, the abbreviation of Nature Catalysis should be given.”

Our Response: Thanks for your suggestions. We have revised the reference in Ref. 47 to use the abbreviation for Nature Catalysis.

7) Comment #7: “In P91 in SI, the integration was in casual version in the ^1H NMR spectra, and the chemical shift was incorrect in the ^{13}C NMR spectra, please standardize the handling of NMR spectra. In P93, 94, 97, 101, 114, 137, 167, 168 in SI, the integration in the ^1H NMR spectra. In P108, 127 in SI, the compounds should be further purified. Please add the compound number in the SI for better reading.”

Our Response: Thanks for your suggestions. We have standardized the handling of the ^1H NMR and ^{13}C NMR spectra for P91, 93, 94, 97, 101, 114, 137, 167, and 168 in the SI. We have further purified the compounds in P108 and P127, and numbered the compounds in the SI. For detailed information, please refer to the Supplementary Information.

4. Responses to reviewer 4:

1) Comment #1: “The abstract could be condensed. My inclination is that the abstract should focus on description of the content of this study (what has been done and what can they demonstrate), and the significance of this study is not necessarily involved in the abstract. Thus the first two sentence “Allylic ethers and alcohols.....Tsuji-Trost reaction” should better be deleted, and detailed discussion on the significance of this work should better be included in the Introduction section.”

Our Response: Thanks for your suggestions. We have removed the first two sentences from the abstract and made corresponding simplifications.

2) Comment #2: “According to the results shown in Table 2, electron-withdrawing substitutions are well tolerated (even preferable) but weak electron-donating alkyl or acyloxy substitutions seem less tolerable (see product 26, 27, 38 and 39). How about strong electron-donating alkoxy (such as p-methoxyphenol), hydroxy (such as quinol) or amino/amido (such as 4-acetamidophenol, a.k.a. Paracetamol, one of the most representative NSAIDs beside Ibuprofen) substitutions? I also wonder why the authors select a model phenol with a long alkyl chain (see product 27). Can this significant C-O coupling takes place with the most typical alkyl phenol, p-cresol?”

Our Response: Thanks for your suggestions. The reaction shows good compatibility with electron-withdrawing substituents, but poor reactivity with electron-donating groups. We speculate that this may be related to the acidity of the phenol substrate. Substrates with strong electron-donating groups, such as alkoxy (e.g., p-methoxyphenol), hydroxy, or amino/amido groups (e.g., acetaminophen), make it difficult to obtain the desired product. For product 27, it was selected from among several substrates with weak electron-donating groups due to its moderate yield, and thus included in the manuscript. The C-O coupling reaction with the typical alkyl phenol p-cresol is feasible, but the reaction efficiency is low, and only 19% of the yield is obtained.

3) Comment #3: “According to the results shown in Table 3, a series of primary alcohol could smoothly react albeit in declined yields. How about secondary alcohols (such as isopropanol) beside HFIP that is too specific?”

Our Response: Thanks for your suggestions. The reactivity of secondary and tertiary alcohols under this catalytic system is not optimal. We have compiled these data and placed them in the

Supplementary Information.

4) Comment #4: “This remote directed ligand exchange strategy to facilitate allylic C-O coupling of unactivated alkenes are very significant and promising. However, with regards to application and utility, the amide linker between olefin and the quinoline directing group might not be the most satisfactory solution. According to the results shown in Figure 3b, the effectiveness for removal of the DG is not very well, since an excessive amount of a relatively expensive oxidant IBX is employed and the 53% yield is too low for utility consideration. It is strongly proposed that, in future studies, the authors develop more applicable linkers other than amide bond, as well as more atom-economical DGs other than quinoline.”

Our Response: Thanks for your suggestions. In future studies, we envision removing the AQ directing group and developing it into a chiral ligand to enable asymmetric functionalization reactions at the allylic position. This research is still ongoing.